# Advances in Genetic Analysis and Breeding of Cassava (*Manihot esculenta* Crantz): A Review

**DOI:** 10.3390/plants11121617

**Published:** 2022-06-20

**Authors:** Assefa B. Amelework, Michael W. Bairu

**Affiliations:** 1Agricultural Research Council, Vegetable and Ornamental Plants, Private Bag X293, Pretoria 0001, South Africa; bairum@arc.agric.za; 2Faculty of Natural & Agricultural Sciences, School of Agricultural Sciences, Food Security and Safety Focus Area, North-West University, Private Bag X2046, Mmabatho 2735, South Africa

**Keywords:** biotic stresses, genetic diversity, genomic tools, *Manihot esculenta*

## Abstract

Cassava (*Manihot esculenta* Crantz) is the sixth most important food crop and consumed by 800 million people worldwide. In Africa, cassava is the second most important food crop after maize and Africa is the worlds’ largest producer. Though cassava is not one of the main commodity crops in South Africa, it is becoming a popular crop among farming communities in frost-free areas, due to its climate-resilient nature. This necessitated the establishment of a multi-disciplinary research program at the Agricultural Research Council of South Africa. The objective of this review is to highlight progress made in cassava breeding and genetic analysis. This review highlights the progress of cassava research worldwide and discusses research findings on yield, quality, and adaptability traits in cassava. It also discusses the limitations and the prospects of the cassava R&D program towards development of the cassava industry in South Africa.

## 1. Introduction

With increasing climate variability, cassava has attained significant importance in worldwide agriculture. It is an attractive food security and commercial crop for subsistence farmers with limited resources and it can grow in marginal soils [1]. Cassava roots are the main carbohydrate storage organs and store up to 85% starch on a dry weight basis [2]. Cassava roots contain a starch content of 40% higher than rice and 25% more than maize [3]. The importance of cassava for food security, climate risk mitigation, import substitution for industrial starch, livestock feed, and biofuel feed stock to South Africa’s economy has been reviewed by Amelework et al. [4].

It is estimated that in South Africa, maize accounts for approximately 95% of the country’s starch production, with 37% of the produce being used for food, 40% for feed, 18% for export, and 5% for starch. Due to drought and competition between industries utilizing maize products, the local starch industries failed to meet the starch demand of the country. Hence, South Africa is importing more than 66,000 tons of starch annually [5]. It was reported that cassava starch is preferred in South Africa in terms of imports and it fetches a higher price on the market than maize, potato, or wheat [4]. Hence, grasping the industrial potential of cassava in the starch industry would satisfy local starch demands, avoid computation among staple food commodities, and reduce import volume.

The Agricultural Research Council, together with the State partners, undertook basic research to determine the suitability of cassava as a food, feed, and industrial crop. The research was undertaken as participatory trials with farmers in KwaZulu-Natal, Limpopo, and Mpumalanga. The knowledge of the farmers about the resilience of the crop and its potential to alleviate their food security problem in these provinces and their surroundings triggered extremely high demand (Unpublished). The objective of this review is to highlight progress made in cassava breeding and genetic analysis. This review discusses research findings focused on yield, quality, and adaptability traits in cassava. The potential of the available genetic and genomic resources for breeding and genetic improvement of cassava for enhanced resistance to biotic stresses have been discussed. Finally, the paper highlighted future research focus areas to effectively integrate and commercialize cassava in South Africa.

## 2. Origin and Domestication of Cassava

Cassava belongs to the family Euphorbiaceae, sub-family Crotonoideae, tribe Manihotae, and genus Manihot. The genus has two sections, the Arborae, containing tree species, and the Fructicosae, comprising slow-growing shrubs adapted to savannah grassland or desert conditions [6]. The genus Manihot is a large family of flowering plants with 300 genera and around 8000 species [7]. Of the many species that belong to the genus Manihot, *Manihot esculenta* Crantz is the most economically important species and widely cultivated for food and industrial applications [8].

Cassava is believed to have been domesticated before 4000 BC, and its center of origin is hypothesized to be South America [9]. The questions about the wild progenitor of cassava, the area where the wild progenitor evolved and was initially cultivated have been obscured. Originally, it was speculated that cassava could be ascended and evolved via periodic introgression of genes involving a number of wild species (compile-species) [10]. However, later it was reported that a wild species, *Manihot flabellifolia* (Pohl), was found to be the closest wild relative to cassava [11]. Many studies have supported the ancestral relationship of the modern cultivated cassava and the wild subspecies *M. esculenta* ssp. *Flabellifolia* [12,13,14,15]. 

The exact geographical origin of the crop has been disputed for many years following the appearance of new evidence. Geographic origin infers the distribution and habitats of the wild cassava species [14]. Although controversial results have been reported, the southern border of the Amazon basin currently stands as the recognized center of origin for *M. esculenta* [8,15,16]. A large number of wild Manihot species have been reported to be found in Brazil. Hence, this region is the likely primary center of diversity of cassava [17]. The domestication process involved selection for root size, growth habit, stem number, and ability of clonal propagation through stem cuttings [18]. During the 16th century, Portuguese traders introduced cassava to Africa and later to Asia [18]. It was initially cultivated in the Democratic Republic of Congo and adopted as a famine-reserve crop. Currently, cassava is cultivated in about 40 African countries, stretching over a wide belt from Madagascar in the Southeast to Cape Verde in the Northwest [19].

## 3. Genetic Diversity in Cassava

Genetic improvement begins with the collection and evaluation of diverse genetic resources [1]. Various evolutionary forces such as mutations, migration, hybridization, and polyploidization are responsible for creating variation in plants [20]. In cassava, it is believed that the wide range of genetic diversity was generated through natural and artificial hybridization between the wild *Manihot* spp. and cultivated cassava or through apomixes [21]. Jennings [22] suggested that the high genetic diversity of cassava has resulted from migration followed by natural hybridization. Fregene et al. [23], on the other hand, indicated that genetic diversity was dictated by specific geographic adaptation, while Asante and Offei [24] argued that geographical region might have little effect on genetic diversity in cassava. 

Currently, several national and international research institutes hold large collections of cassava germplasm. The International Centre for Tropical agriculture (CIAT) in Colombia comprises more than 6000 cassava accessions collected mainly (97%) from Latin America and of which 30% of the accessions are of Brazilian origin [25]. The International Institute for Tropical Agriculture (IITA) in Nigeria, on the other hand, consists of more than 2000 genotypes largely of West African origin [26]. Understanding the nature and magnitude of genetic variation present within and among individuals, populations, species, and gene pools is crucial for the efficient management of genetic resources [27]. Genetic diversity is assessed with morphological, biochemical, and/or molecular markers. However, the environment and genotype by environment interaction effects limit the effectiveness of genetic diversity studies using morphological and biochemical markers [28]. Molecular markers permit the detection of genetic differences among closely related and wild species. Various DNA marker systems have been developed and utilized to assess the genetic diversity of cassava germplasm (Table 1).

The success of genetic improvement of any trait depends on the nature of variability present within the trait [52]. In modern cassava breeding, a source population with high frequencies of alleles associated with desirable characters are developed through collection or hybridization of genotypes derived from selected elite clones [1]. New improved cultivars can be developed following selection of superior clones from segregating populations [53].

## 4. Breeding for Yield and Quality Related Traits

### 4.1. Starch Content

Starch has been the subject of intensive research over many decades. Cassava is the second most important source of starch worldwide, after maize, and the most traded one [54]. The average starch content of cassava is 84.5% on a dry weight basis and the average amylose content is 20.7%, ranging between 15.2 and 26.5% [55]. Cassava starch is the cheapest and the most preferred known form of starch with many positive characteristics such as high paste clarity, relatively good stability to retrogradation and swelling capacity, low protein complex, and good texture [56]. Novel starch properties such as amylose-free and high-amylose starch are of interest to the cassava community [57]. 

South Africa is a highly industrialized country and starch is widely used by the food, alcohol, textile, pharmaceutical, cosmetic, adhesive, paper, and plywood industries. The Republic is Africa’s largest producer of starch and exports starch to neighboring countries. The starch industries in South Africa have been producing starch mainly (95%) from maize. There is a huge competition among industries utilizing maize products and other uses. Hence, the starch industry is not able to satisfy the local demands. Hence, South Africa has been importing thousands of tons of starch from Southeast Asia annually [5]. Exploitation of cassava as a source of starch will provide an alternative to the maize industry. Producing starch from cassava locally will satisfy local starch demands, avoid competition among staple food commodities, relieve the country’s economy from foreign currency strains, and reduce import volumes [4].

### 4.2. Dry Matter Content

Dry matter content is an important character for the acceptance of cassava by producer, consumer, and processor. It is considered as the economic or true biological yield, which is controlled by polygenes [58]. The proportion of dry matter in cassava storage roots ranges from 17 to 47%, with the majority of the accessions lying between 20% and 40% [59] and values above 30% are considered to be high. Dry matter content is highly influenced by a number of genetic and environmental factors such as age of the crop, efficiency of the canopy to trap sunlight, season, and location effects [56]. The longer the plants stay on the field, the more dry matter and starch is accumulated. It was reported that yields of fresh and dry roots as well as starch increased progressively from 8 to 18 months after planting (MAP).

It was reported that, on average, nearly 90% of cassava root dry weight is carbohydrate, 4% crude fiber, 3% ash, 2% crude protein, and 1% fat [60]. Dry matter partitioning into different plant parts varies with the growth cycle of the plant. During the early growth stages of the plant, more dry matter is accumulated in the leaves than in the stems and storage roots [61]. However, at 4 MAP about 50–60% of the total dry matter is accumulated in the storage roots [61]. 

Total dry weight of a cassava plant is positively associated with dry weight of the storage roots, suggesting that both traits could be improved simultaneously [62]. However, dry matter content is not associated with fresh root yield. Okogbenin et al. [63] reported that dry matter content is positively and highly correlated with starch content and harvest index, suggesting the importance of the two traits for indirect selection. Adjebeng-Danquah et al. [64] also reported a significant positive correlation between early storage root yield and final root yield, indicating the possibility of selecting genotypes with high yield potential at 6 MAP. Under tropical conditions, maximum rate of dry matter accumulation was detected at 3–5 MAP [65], while at high attitudes maximum rate was observed at 7 MAP [66]. Similarly, Kawano et al. [67] reported that root dry matter content tended to be higher at 8 MAP compared to 12 MAP. Although the rate of dry matter accumulation varies with the genotypes and growing conditions, it was suggested that selection for dry matter content should not be performed before 4 MAP [61]. 

### 4.3. Early Bulking

Early bulking is an important trait in cassava, referring to the thickening of storage roots as they fill with assimilates after the plant satisfies its need for vegetative growth [63]. Early bulkers are those cultivars that are harvestable within 7–8 MAP. Breeders have been used early bulking as a drought tolerance mechanism [64]. Various researchers have different opinions on the onset and rate of bulking. Doku [68] suggested that cassava plants begin root bulking at 2 MAP but reasonable storage root mass was not reached until 6 MAP. However, according to El-Sharkawy [69], cassava root bulking starts at 3 MAP but roots become a major sink between 6 and 10 MAP. Similarly, Izumi et al. [70] reported that cassava root bulking starts at 3 MAP but rapid starch deposition does not occur before 6 MAP. Wholey and Cock [71] argued that although root bulking starts at 2 MAP, the rate of root bulking increased with time and varied with cultivars. It was suggested that earliness in cassava is associated with either early onset or rapid rate of bulking or a combination of the two [71]. Various researchers who studied the rate of accumulation of storage root mass at different harvesting times indicated the existence of early bulking genotypes [63]. It was also reported that cultivars showing a high bulking rate over a long period of time produced more storage root yield than those with intermediate or low bulking rates for a short duration [64]. Suja et al. [72] also indicated that short duration cultivars exhibit the maximum bulking rate during their early growing stage.

Cassava, being a tropical crop, prefers humid-warm climates with temperature ranges of 25–29 °C and an altitude below 1500 m. Cassava is highly sensitive to low temperatures below 18 °C [73]. Low temperature causes delayed sprouting of stem cuttings, reduced leaf expansion, low biomass accumulation, and decreased storage root yield [74]. In South Africa, the growing season is characterized by a hot rainy summer followed by a cold and dry winter. Frost is a major obstacle for cassava production and propagation in some parts of South Africa. 

The most suitable areas for cassava production are northern KwaZulu-Natal, the eastern parts of the Limpopo and Mpumalanga. Due to frost, the vast majority of the Eastern Cape, Western Cape, Northern Cape, and the Free State are not suitable for cassava cultivation. The frost days ranged from 30 days in the Eastern Cape to 120 days in the Free State and the Northern Cape. Successful cultivation of cassava throughout the republic requires cold tolerance and short growth cycle cultivars. All the cultivars currently in the system take more than 18 months to mature and are highly sensitive to cold. Therefore, early bulking cultivars, that fit into the growing season (i.e., matures within 7–9 MAP), or cold-tolerant cultivars that can grow in a prolonged growth period are in demand. Early maturing cassava genotypes that can escape the cold winter should be the focus of cassava breeding in South Africa.

### 4.4. Cyanogenic Content

Cyanogenic glycosides (CN) are a group of chemical compounds that produced hydrogen cyanide following enzymatic breakdown [75]. There are at least 25 cyanogenic glycosides known to be found in the edible parts of plants [76]. Many species produce and sequester cyanogenic glycosides including cassava, sorghum, almonds, lima beans, flax, and white clover [77]. In plants, glycoside hydrolysis occurs when the plant tissues have been disrupted by herbivores, fungal attack, or mechanical damage. Although many explanations have been given on the importance of CN in plants, the most probable physiological role of CN is defense against herbivores, pathogens, competitors, and theft [78]. In support of this view, bitter cassava genotypes exhibited greater tolerance to CMD [79] and drought than sweet cassava. 

The predominant cyanogenic compounds in cassava roots and leaves are linamarin (95%) and lotaustralin (5%) [80]. Depending on the genetic, physiological, climatic, and edaphic factors, the normal range of cyanogenic glycoside content in cassava ranged from 1 to 1300 mg per kg of dry weight [81]. However, some reports indicated that the total cyanogenic content of the roots was not correlated with the content in the leaves and stems of the same plant [82]. The levels of cyanogenic glycosides in cassava roots are generally lower than that in the leaves and stems [83]. It has been reported that cassava roots contain a cyanide content of 10–500 mg per kg of dry matter [84], while the leaves contain 53–1300 mg per kg of dry matter [85].

Cassava cultivars have been classified biochemically using cyanogenic glucoside content as bitter and sweet [10] depending on the presence or absence of toxic levels of cyanogenic glucosides, respectively. The bitter cultivars are characterized by their high cyanogenic content (100–400 mg per kilogram of fresh weight of roots) distributed throughout the storage roots, whereas those cultivars with very low cyanogenic content (15–50 mg per kilogram of fresh weight of roots) mainly confined in the peel are termed as sweet varieties [76]. The utilization of cassava as food and feed is limited by the toxic level of cyanogen. Consumption of raw or inadequately processed cassava can lead to chronic and acute health problems resulting from cyanide poisoning [78]. The World Health Organization (WHO) recommended 10 ppm or 10 mg hydrogen cyanide per kg of cassava flour as a safe level [86]. Sweet cassava roots can be eaten by peeling and cooking, whereas bitter varieties require more extensive processing such as peeling, washing, grating, fermenting, drying, or frying. Efforts have been made on breeding and selection of low-cyanogen varieties [80], development of low-cyanogen mutants through mutagenesis [87], and genetic engineering [88]. 

### 4.5. Post-Harvest Physiological Deterioration (PPD)

PPD is a major constraint for the production, development, expansion, and exploitation of cassava as an industrial crop in many parts of the world [89]. It is the outcome of intricate interactions of simultaneously occurring cellular functions in the harvested cassava roots and results in considerable quantitative and qualitative post-harvest losses of the fresh cassava roots. Saravanan et al. [90] reported that the estimated yield losses of fresh cassava roots due to PPD are nearly a third of the total harvest worldwide. The expression of PPD is controlled by genotypic and environmental factors following microbial infections [91]. In Thailand where cassava is the most important industrial crop, PPD results in an economic loss of up to USD 35 million annually [92]. Rudi et al. [93] predicted that extending the shelf life of cassava to several weeks would reduce financial losses by USD 2.9 billion in Nigeria over a 20-year period. Studies indicated the presence of genetic variation for PPD among genotypes [94]. However, environmental factors such as age of the plant, root conditions during harvest and thereafter, and storage conditions significantly influence the development and effects of PPD [95].

## 5. Breeding for Biotic Stresses

### 5.1. Cassava Mosaic Disease (CMD)

Cassava production in Africa is curtailed by cassava mosaic disease (CMD) and cassava brown streak disease (CBSD) [96]. CMD is a severe cassava disease prevalent in all cassava growing regions of Africa and India [97]. However, variation in overall prevalence and in the severity of losses caused by the disease has been reported among regions [98]. CMD is caused by a complex of diverse whitefly-transmitted cassava mosaic geminiviruses (CMGs) [99]. The cassava geminivirus family is composed of at least 11 distinct viruses that have been characterized worldwide, of which seven have an African origin [100]. The African viruses include African cassava mosaic virus (ACMV) [101], East African cassava mosaic virus (EACMV) [102], East African cassava mosaic Malawi virus (EACMMV) [103], East African cassava mosaic Cameroon virus (EACMCV) [104], South African cassava mosaic virus (SACMV) [105], East African cassava mosaic Zanzibar virus (EACMZV) [106], and East African cassava mosaic Kenya virus (EACMKV) [107]. Each of the cassava mosaic geminiviruses (CMGs) can cause CMD and the virus combinations are more detrimental than single infections [108]. Generally, in Africa the estimated yield losses caused by CMD were reported at 15–24%, representing 15–28 million tonnes of cassava production [109]. However, in a recent report by Tembo [110], the reduction in yield can be more than 70%. The estimated annual economic losses in East and Central Africa are estimated to be between USD 1.9 and 2.7 billion [99]. 

The mechanism of resistance to CMD is not fully understood and no immunity to CMD has been reported within the cultivated species. The mechanism of inheritance to CMD was previously thought only to be polygenic. However, three sources of host-plant resistance are known and currently exploited by breeders to combat CMD. The first source of resistance was introgressed from *Manihot glaziovii* [111] and the mechanism was later found to be polygenic recessive [112]. The polygenic nature of this resistance is often goes with small race-specific effects that have been reported in cassava. The gene conferring this resistance was mapped and named as *CMD1* [113]. The first microsatellite markers to be associated with *CMD1* locus were reported by Fregene and Puonti-Kaerlas [113]. Later, Akano et al. [114] confirmed that the *CMD1* gene was associated with the *SSY40* marker on linkage group D on the TMS 30572 derived genetic map. 

The second source of CMD resistance was discovered in a Nigerian landrace called TME 3 (Tropical *Manihot esculenta*) [115]. The resistance mechanism of TME 3 was controlled by a single dominant gene, named *CMD2* [114]. Using bulk segregant analysis, Akano et al. [114] identified two markers, *SSRY28* and *GY1*, flanking the *CMD2* locus at distances of 9 and 8 cM, respectively. At least five markers tightly associated to *CMD2* have been developed, the closest being *RME1* and *NS158*, at distances of 4 and 7 cM, respectively [114,115,116,117,118].

The third source of resistance identified in cultivar TMS 97/2205 was reported to be a quantitative trait locus (QTL), named *CMD3* [119]. Unlike *CMD1* and *CMD2*-type resistances, for plants infected with cassava mosaic graminviruses (CMGs) *CMD3* confers very high levels of resistance to CMD with little or no expression of the disease on the leaves [115]. 

### 5.2. Cassava Brown Streak Disease (CBSD)

Cassava brown streak disease (CBSD) is caused by cassava brown streak virus, which recently was taxonomically grouped in the family Potyviridae and genus *Ipomovirus* [120,121]. The disease was first reported in Tanzania in the early 1930s [122], where it was restricted to the lowland coastal areas of eastern Africa [123]. Recent surveys have shown that CBSD is highly prevalent in Central, Eastern, and Southern parts of Africa [124]. It has been reported in Mozambique, Kenya, Uganda, Zambia, and Malawi [124]. The name brown streak was given to the disease due to the brown lesions symptom that appears on the young stems. The viral RNA genome sequences sampled from CBSD-diseased plants across eastern and southern Africa revealed the presence of two distinct virus species, namely, Uganda cassava brown streak virus (UCBSV) and cassava brown streak virus (CBSV) [125]. 

CBSD does not have an obvious effect on the growth of cassava; however, the root necrosis produced by CBSD has caused a reduction in both qualitative and quantitative yield [126] and affects maintenance of planting materials [127]. Most of the yield loss from the disease is thought to be the consequence of the loss of root storability resulting from severe root rot [128]. Gondwe et al. [129] reported 18–25% yield loss by CBSD, while Hillocks et al. [128] published a yield loss estimate of 70% from the most susceptible variety. Much less attention has been given to the disease compared to CMD, partly due to its restricted geographic distribution. However, recently, the high prevalence and distribution of the disease has been reported due to the presence of a large population of whitefly vector *B. tabaci*. CBSD has not been reported in South Africa, but the disease is highly prevalent in Mozambique, Malawi, Zimbabwe, and Zambia [130]. With the recognition of the threat posed by CBSD to food security in the neighboring countries, breeding for CBSD is a priority for Eastern and Southern Africa cassava improvement programs. Cassava genotypes with resistance to the two-virus families: *geminiviridae* and *potyviridae* have been identified. These genotypes are useful genetic resource for advancing disease resistance in cassava (Table 2).

## 6. Genomic Resources of Cassava

Cassava is a naturally diploid species (2n = 36) with a genome size of 770 Mbp [149]. Although most of the species studied contain 36 chromosomes and are regarded as diploids (2n), irregular pairing at meiosis and spontaneous polyploidization (e.g., triploids (3n) and tetraploids (4n)) have been reported in both wild and domesticated cassava [150]. Likewise, the presence of three nucleolar chromosomes has been reported in cassava, which is high for true diploids, suggesting that Manihot species are probably segmental allotetraploids with a basic chromosome number x = 9 [6].

The draft cassava genome (draft v4.1) was released in 2009 [151] and is publicly available via the V10 Phytozome platform [152]. The cassava genome was constructed using the whole genome shotgun approach and assembled into 12,977 scaffolds spanning a total of 532.5 Mb [151]. However, the initial assembly only covered 69% of the estimated cassava genome based on nuclear DNA quantity [153]. Consequently, post-genome improvement efforts have been underway to build upon the genome data of cassava. Recently, Sato and his co-workers anchored 189 new scaffolds to the genetic map leading to an extension of the present cassava physical map by 30.7 Mb [154]. ICGMC [155] improved the v4 assembly by enriching it into chromosome-scale and by ordering and orienting 57% of the v4 assembled sequences into 18 chromosomes. Then, Bredeson et al. [156] advanced v5 to v5 by capturing 18% more total contig sequence, increasing overall sequence contiguity, and incorporating 45% more sequence into chromosomal scaffolds. The v7 reference genome assembly followed the same strategy as v6, but with more contiguous underlying sequences assembled de novo from Pacific Biosciences (PacBio) single-molecule real-time (SMRT) continuous long-read (CLR) data. The v7 reference assembly had 669 Mbp, which is somewhat shorter than the estimated 750 Mbp haploid genome size [157]. The reference genome has served as a platform for studying the gene content, gene expression, and genetic variation for genomic analyses since its release [158,159,160,161,162,163,164]. Due to the highly heterozygous nature of the crop, high repetitive sequences, often transposon-related, are interspersed among the compact genes of cassava [151]. Despite the continuous efforts of resolving the gaps and haplotype problem, the currently available cassava genome is not complete. Recently, Qi et al. [165] reported the most accurate, continuous, complete, and haplotype-resolved cassava genome assembly. Further genomic studies through genome-wide identification of sequences that encode traits of economic value and mapping them on the genome are needed to accelerate genome-wide genetic improvements. 

cDNA libraries have been constructed for varieties selected for starch content and bacterial blight resistance [166], virus resistance [167], abiotic stress tolerance [168], carbohydrate structure [169] and content, and β-carotene content and composition [169].

In the absence of a perfect and complete reference genome, the development of a genetic linkage map remains a necessary prerequisite for effective marker-assisted selection. Many genetic linkage maps have been developed for cassava. The first molecular genetic map was generated with 132 RFLP, 30 RAPD, 3 SSR, and 3 isozymes markers using 90 F1 plants segregating from an intraspecific cross between TMS 30572 and CM 1777-2 [170]. The map consists of 20 linkage groups spanning 931.6 cM with an average marker density of 1 marker per 7.9 cM. Later in 2001, an additional 36 SSR markers were integrated into this map [171]. A second map was constructed using 268 individuals derived from selfing accession K150 [172]. Okogbenin et al. [172] initially screened a total of 472 SSR markers on K150 and its parents TMS 30572 and CM 2177-2. The result revealed that only 122 SSRs were found to be polymorphic. The map was constructed using 100 polymorphic SSRs spanning 1236.7 cM, distributed across 22 linkage groups with an average marker distance of 17.92 cM. 

Kunkeaw et al. [173] reported another genetic linkage map constructed with 119 AFLPs and 18 SSRs, spanning 1095 cM with an average of 7.99 cM between markers. Later in 2011, a linkage map that consisted of 510 markers covering 1,420.3 cM, distributed over 23 linkage groups with a mean distance between markers of 4.54 cM was developed [174]. Sraphet and his co-workers primarily used 640 new SSR markers, from an enriched genomic DNA library of the cassava variety Huay Bong 60 and 1500 novel EST-SSR loci from the Genbank database. Rabbi et al. [175] reported a genome-wide SNP and SSR-based genetic map of cassava constructed from an F1 population derived from a cross between Namikonga and Albert. This map resulted in a final linkage map of 1837 cM containing 568 markers (434 SNPs and 134 SSRs) distributed across 19 linkage groups with average inter-marker distance of 3.4 cM [175]. More recently, a highly dense genetic map was developed using genotype-by-sequencing (GBS) based SNP markers for mapping the cassava mosaic disease resistance gene [148]. Efforts in understanding the structure and function of the cassava genome have been in progress. To this effect, several transcriptome and proteomics tools have been developed and several cassava ESTs and full-length cDNA sequences were generated and characterized [148,150]. Furthermore, proteomics studies have contributed to the identification of several cassava proteins involved in many biological processes such as growth, development, and response to abiotic and biotic factors [176].

Molecular markers are playing a significant role in genome mapping, gene tagging, genetic diversity, and phylogenetic analysis. In cassava, since the beginning of the 1990s, molecular and genomic tools have been developed to elucidate the genetics of the traits of economic importance. Various molecular tools have been developed in cassava, which include random amplified polymorphic DNA (RAPD) [177], amplified fragment length polymorphism (AFLP) [178], restriction fragment length polymorphism (RFLP) [179], simple sequence repeats (SSR) [173,174], and single nucleotide polymorphism (SNP) [180,181]. Recently, the generation Challenge Program (GCP) converted 1740 SNPs in cassava for use on the KASPar platform (LGC) [182]. The availability of genomic tools is indispensable for the utilization of the tools in cassava breeding. In cassava, indirect selection of traits using molecular markers has been implemented for different traits of economic importance. Marker-assisted selection using molecular markers linked to resistance to CMD [114,116,183], CBSD [161], cassava anthracnose disease [184], cassava green mite (CGM) [185], cassava bacterial blight (CBB) [186,187], beta-carotene content [188], early root yield [189], dry matter and starch content [162] have been practiced. 

## 7. Outlook for Future Research on Cassava Breeding and Genetic Improvement

The current challenge posed by climate change calls for more sustainable options. According to Davis-Reddy and Vincent [190], the northeastern and the eastern parts of the countries where the majority of the agricultural production takes place showed high climate vulnerability indexes. Cassava is one of the most resilient crops that offers immense opportunity as a food, feed, and industrial crop.

The production, characterization, and product development from cassava is at its infancy in South Africa compared to other African countries. Although cassava is not one of the traditional commodity crops in South Africa, farmers have grown it as a minor crop in the far northeast sector of the country. There is a need to establish a sustainable genetic resource management system that focuses on the characterization, conservation, and utilization of the broad cassava genetic base. It is also wise to exploit the available genetic and genomic resources that have been developed in many national and international research organizations. 

Significant industrial opportunities (starch, animal feeding, or ethanol production) exist for the production and use of cassava in South Africa. If the crop improvement and germplasm development activities are dictated by the target market, the impact and adoption of cassava will be fast and significant [191]. Understanding of the role of the different stakeholders in the value-chain and analyzing the relative importance of each market segment and the varietal requirements of each market are important to properly satisfy the expectations and the demands [192].

Applied research can facilitate agricultural transformation by developing varieties with high yield, better quality, better environmental adaptability, good agronomic practices, and appropriate mechanization technologies. In cassava, due to its high heterozygosity nature, long breeding cycle, and low multiplication rate, breeding advances have not been as impressive as those reported in cereals. Recently, advances in breeding approaches have been developed and reported for cassava [191,192,193,194]. These approaches will be implemented for breeding for yield, better quality, and disease resistance in cassava. Policies should foster public–private partnerships for technology development and link them to markets in order to facilitate the scaling up of successful innovations. 

Market creation and product diversification should be important to assimilate cassava in the present production system and facilitate cassava commercialization and agro-processing. In South Africa, the agricultural produce market is largely controlled by corporate companies with excessive regulatory and compliance requirements. This can only be ameliorated by placing enabling policies and identifying key institutions for sustainable cassava value chain and commodity organization development. Although the South Africa cassava industry association (CIASA) was established under the Department of Trade and Industry (DTI), currently the association is not active. CIASA should be revived by revisiting the constitution and the members and its role should be redefined to support the cassava industry in South Africa and to include all the value chain actors. 

One of the constraints hampering the exploitation and diffusion of cassava in South Africa is the lack of awareness and lack of public and private sector investment. Exploitation of global market opportunity requires public–private partnerships. Policies should foster public–private partnerships for technology development, agro-processing, and marketing of cassava to facilitate the scaling up of successful innovations.

In most of the African countries, the supply chain for cassava products begins with small-scale production units, followed by small-scale processing units for the drying and/or milling of cassava. These steps are often carried out at the home and village/local level. As the market grows, the supply chain activities such as marketing, processing, and packaging are undertaken by fewer larger-scale units, which then distribute the final product to a larger number of consumers. This hourglass supply chain differs from the supply chain of many established agricultural products. The existence of the hourglass supply chain does suggest that the growth and development of cassava product markets will benefit the large number of resource-poor farmers located on poor lands as well as the local processing units. 

The challenge is how to equip these farmers and processors with the knowledge and tools needed to provide the products that meet the requirements of growth markets. Training is an integral part of any development activity and a process of acquiring new knowledge, skills, practices, and attitude in the context of preparing farmers for improving agricultural productivity. Training plays a key role in human capacity development, to equip farmers with skills, knowledge, and competencies for sustainable crop production, resource utilization, and income generation. 

The development of the cassava industry can contribute to food and income security, job creation, and revitalization of the rural sector. Public investment in cassava R&D and product development played a vital role in reducing food insecurity, malnutrition, unemployment, and urban migration. 

## 8. Conclusions

Cassava can be used as an alternative food security and industrial crop in South Africa. It can grow and produce reasonable yields in areas where cereals and other crops are not viable. It can tolerate drought and can be grown on soils with low soil fertility but responds well to irrigation and fertilizers. However, it has received little research priority and a limited number of genotypes have been tested. The currently available wide genetic resources maintained in international and regional research institutes should be explored for traits such as quality, yield potential, and biotic and abiotic stress tolerance. These will identify promising genotypes, which may facilitate development of genetically superior and improved cassava cultivars for South Africa. In addition, exploiting the industrial potential of cassava in South Africa will improve rural livelihoods through income generation and job creation. Furthermore, the national economy should benefit indirectly from job creation, and directly from foreign exchange savings originating from replacing imported products and raw materials.

## Figures and Tables

**Table 1 plants-11-01617-t001:** Genetic diversity analyses studies reported in cassava.

Marker System	No Acc. Evaluated	Species	References
RFLP	80	*M. esculenta*, *M. glaziovii*, *M. Caerulescens*	[29]
AFLP	105	*M. esculenta* ssp. *esculenta*, *M. esculenta* ssp. *Flabellifolia* *M. aesculifolia*, *M. carthaginensis*, *M. tristis*, *M. brachyloba*	[12]
RAPD	31	Cultivated cassava	[30]
RAPD/AFLP	53	*M. esculenta*, *M. flabellifolia*, *M. peruviana, M. glaziovii*, *M. reptans*, *M. chlorostica*, *M. aesculifolia*, *M. michaelis*	[31]
AFLP	76	Cultivated cassava	[32]
SSR	212	*M. esculenta* ssp. *flabellifolia*, *M. pruinosa*	[15]
RAPD	24	*M. esculenta* ssp. *esculenta*	[33]
Isozyme	46	Cultivated cassava	[34]
SSR	117	Cultivated cassava	[35]
SSR	185	Cultivated cassava	[36]
SSR	37	Cultivated cassava	[37]
SSR	55	Provitamin-A cassava	[38]
SSR	54	Cultivated cassava	[39]
SSR	1401	*M. esculenta*	[40]
SSR	163	*M. esculenta*	[41]
ISSR	17	Landraces	[42]
SSR	596	*M. esculenta* ssp. *esculenta*, *M. esculenta* ssp. *flabellifolia*	[43]
SNP	1580	Cultivated and wild cassava	[44]
SNP	3000	Cultivated cassava	[45]
SSR	120	*M. esculenta*	[46]
SSR	157	Cultivated cassava	[47]
SNP	183	Provitamin-A cassava	[48]
SNP	105	Cultivated cassava	[49]
SSR	89	Cultivated cassava	[50]
SNP	102	Cultivated cassava	[51]

**Table 2 plants-11-01617-t002:** Disease resistance reported in selected *Manihot esculenta* and related cassava species.

No Acc. Evaluated	Accession	Origin	Stress Tolerance	References
180	58308	East Africa	CMD	[131]
40	TME3, TME4, TME5, TME6, TME7, TME9, TME11, TME13, TME28	Nigeria	CMD	[23]
150	TMS30572	IITA	CMD	[132]
158	TME 3	Nigeria	CMD	[114]
65	TME7	Nigeria	CMD	[116]
40	TMS 98/0581, TMS 99/3073, TMS 97/4763, TMS M98/0040, TMS 98/0505, TMS 97/0211, TMS 97/4769, TMS 99/2123, TMS M98/0068, TMS 97/0162	West Africa	CMD	[133]
156	CR 14A-1, AR-38-3, CR41-10	Latin America	CMD	[118]
21	11Q, T7, N13	Asia	CMD	[134]
116	NASE 1, MM96/4271, CR 20A-1, TZ06/130, MM96/0686, MM96/0876	East Africa	CBSD	[135]
5	Namikonga	East Africa	CBSD	[136]
200	TMS 97/2205	Nigeria	CMD	[119]
19	TME 11	Ghana	CMD	[137]
14	TMSI 95/0528, TMSI 95/0211, 96/1089A, TMS 92/297, TMS 83/138, TMS 91/377, MV 99/0038	Congo	CMD	[138]
38	Capevars	Ghana	CMD	[139]
11	AR15-5, CR52A-4, CR52A-25, CR42-4, CR59-4, AR14-10, CR41-10, CR52A-31	CIAT	CMD	[140]
14	KBH2006/18	Africa	CBSD	[141]
10	TMS 30572, TME 3, TME 204, Oko-iyawo, TMS 97/0505	West Africa	CMD	[100]
238	DSC118, DSC167	South America	CBSD	[142]
6	MM2006/128, MM2006/123, MM2006/130	East Africa	CBSD	[143]
220	NAROCASS 1, NAROCASS 2	East Africa	CBSD, CMD	[144]
24	TME7, TMS01/0098, Atinwewe, BEN/86052, Excel, TMS01/1086052	West Africa	CMD	[145]
14	F10_30R2, Eyope, Mkumba, Mkuranga1, Narocass1, Nase3, and Orera	East and South Africa	CBSD	[146]
102	CR63, KM57	CIAT	CMD	[147]
180	IITA-TMS-011412	IITA	CMD	[148]

## Data Availability

Data is contained within the article.

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
