# Peer review of "Advances in Genetic Analysis and Breeding of Cassava (Manihot esculenta Crantz): A Review"

_plants, 2022, doi:10.3390/plants11121617_

Round 1

Reviewer 1 Report

Cassava roots are rich source for starch and have been consumed by people in many countries as a staple food. In this paper, authors review current status and prospects on genetic analysis and breeding of this important crop. Since first draft genome of cassava that was sequenced by Roche 454 platform was reported in 2009, diverse efforts for genomics-assisted breeding have been performed by many research groups, including high-quality genome sequencing and structural analysis, transcriptome, and high-density genetic maps. However, authors are missing the most recent advances in these genetic studies. Unfortunately, I do not recommend this manuscript to be published on the journal due to the forementioned reason.

Author Response

Response to Reviewer 1 Comments

  1. Cassava roots are a rich source of starch and have been consumed by people in many countries as a staple food. In this paper, the authors review the current status and prospects for genetic analysis and breeding of this important crop. Since the first draft genome of cassava that was sequenced by the Roche 454 platform was reported in 2009, diverse efforts for genomics-assisted breeding have been performed by many research groups, including high-quality genome sequencing and structural analysis, transcriptome, and high-density genetic maps. However, the authors are missing the most recent advances in these genetic studies. Unfortunately, I do not recommend this manuscript to be published in the journal due to the aforementioned reason.

Response 1: Articles that are more recent are added to the manuscript on lines 334-346

Reviewer 2 Report

The authors cite a survey, but no results are shown, or reference.

What is the area under cassava in South Africa?

To start promoting a crop, you don't need a very sophisticated breeding program, introductions from homologous regions can serve the purpose. Market development would be key.

Winter will be a key constraint, particularly for planting material preservation, as well as effective length of the season which in turn will impact production.

Below are a couple of recent reviews done by cassava breeders which represent the most updated information on cassava breeding.

Theoretical and Applied Genetics (2021) 134:2335–2353
https://doi.org/10.1007/s00122-021-03852-9

Crop Breed Genet Genom. 2020;2(2):e200008. https://doi.org/10.20900/cbgg20200008 

Author Response

Response to Reviewer 2 Comments

  1. The authors cite a survey, but no results are shown, or reference.

Response 1: This question is not specific and clear. There is one article reported in this manuscript that is on toxicology survey on plant species that produce and sequester cyanogenic glycosides. 

  1. What is the area under cassava in South Africa?

Response 2: There are no clear figures for South Africa because it has not been adopted as a crop in South Africa on a consistent basis, either by small-scale farmers or large-scale commercial farmers. Historically, small-scale farmers have grown cassava as a minor crop in the far north-eastern part of the country. However, there is an initiative to scale up cassava production, with two discrete areas of interest: large-scale production for industrial starch, and expanding its footprint as a food security crop for small-scale farmers, especially in the context of climate change.

  1. To start promoting a crop, you don't need a very sophisticated breeding program, introductions from homologous regions can serve the purpose. Market development would be key.

Response 3: Cassava is not widely cultivated in South Africa and there might not be big demands. However, the demand did not create the iPhone, the person who invent iPhone create the demand. There are many cassava products imported to South Africa, indicating the existence of few markets for cassava.  There is an existing market for cassava starch but we need to create demand for food cassava to diversify the food base.

  1. Winter will be a key constraint, particularly for planting material preservation, as well as effective length of the season which in turn will impact production.

Response 4: South Africa has both tropical and temperate ecologies. The most suitable areas for cassava production are northern KwaZulu-Natal, the eastern parts of the Limpopo and Mpumalanga. Due to frost, the vast majority of the Eastern Cape, Western Cape, Northern Cape and the Free State are not suitable for cassava cultivation. The frost days ranged from 30 days in the Eastern Cape to 120 days in the Free State and Northern Cape. Successful cultivation of cassava throughout the republic requires cold tolerance and short growth cycle cultivars.

  1. Below are a couple of recent reviews done by cassava breeders which represent the most updated information on cassava breeding. Theoretical and Applied Genetics (2021) 134:2335–2353
    https://doi.org/10.1007/s00122-021-03852-9 and Crop Breed Genet Genom. 2020;2(2):e200008. https://doi.org/10.20900/cbgg20200008 

Response 5: Both articles are included in the manuscript on lines 416-422.  

Round 2

Reviewer 1 Report

Authors improved the manuscript as I commented. However, authors are still missing an important paper as follows; Qi et al (2022) The haplotype-resolved chromosome pairs of a heterozygous diploid African cassava cultivar reveal novel pan-genome and allele-specific transcriptome features. GigaScience 11. Please insert the content of this paper into the manuscript to be revised. The scientific name of cassava in the article title must be amended to Manihot esculenta Crantz.

Author Response

Response: The contents of Qi et al. (2022) are included in the MS on lines 346-349. 

Reviewer 2 Report

The survey I am referring to is the one mentioned on lines 40-51, without any single reference that can lead a person reading this report to take a look at it.

Author Response

Response: The sentence stated on lines 40-45 emphasizes the participatory agronomic research that has been done by the ARC for the last three years. The project is an ongoing project that has not yet been finalized and published. The sentences on lines 41 - 51 talk about the objectives and the significance of this article.

Round 3

Reviewer 1 Report

Authors improved the manuscript as I commented. The scientific name of cassava in the article title must be amended to Manihot esculenta Crantz prior to the final publication.

Author Response

Thanks for the comment, it was a foolish mistake 

The scientific name is corrected 
